# Improved bacterial leaf blight disease resistance in the major elite Vietnamese rice cultivar TBR225 via editing of the *OsSWEET14* promoter

**Phuong Nguyen Duy**[1☯], **Dai Tran Lan**[1,2☯], **Hang Pham Thu**[1], **Huong Phung Thi Thu**[1], **Ha Nguyen Thanh**[1], **Ngoc Phuong Pham**[1¤], **Florence Auguy**[3], **Huong Bui Thi Thu**[4], **Tran Bao Manh**[5], **Sebastien Cunnac**[3], **Xuan Hoi Pham**[1]*

**1** Department of Molecular Pathology, Institute of Agricultural Genetics, Vietnam Academy of Agricultural Sciences, Hanoi, Vietnam, **2** Faculty of Natural Sciences, Department of Applied Biology and Agriculture, Quynhon University, Quynhon, Vietnam, **3** PHIM Plant Health Institute, Univ Montpellier, IRD, CIRAD, INRAE, Institut Agro, Montpellier, France, **4** Vietnam National University of Agriculture, Hanoi, Vietnam, **5** ThaiBinh Seed Corporation, Thaibinh, Vietnam

☯ These authors contributed equally to this work.
¤ Current address: Faculty of Biology, Hanoi University of Sciences, Hanoi, Vietnam
* xuanhoi.pham@gmail.com

**Data Availability Statement:** All relevant data are within the paper and its Supporting Information files.

## Abstract

TBR225 is one of the most popular commercial rice varieties in Northern Vietnam. However, this variety is highly susceptible to bacterial leaf blight (BLB), a disease caused by *Xanthomonas oryzae* pv. *oryzae* (*Xoo*) which can lead to important yield losses. *OsSWEET14* belongs to the *SWEET* gene family that encodes sugar transporters. Together with other Clade III members, it behaves as a susceptibility (*S*) gene whose induction by Asian *Xoo* Transcription-Activator-Like Effectors (TALEs) is absolutely necessary for disease. In this study, we sought to introduce BLB resistance in the TBR225 elite variety. First, two Vietnamese *Xoo* strains were shown to up-regulate *OsSWEET14* upon TBR225 infection. To investigate if this induction is connected with disease susceptibility, nine TBR225 mutant lines with mutations in the AvrXa7, PthXo3 or TalF TALEs DNA target sequences of the *OsSWEET14* promoter were obtained using the CRISPR/Cas9 editing system. Genotyping analysis of $T_0$ and $T_1$ individuals showed that mutations were stably inherited. None of the examined agronomic traits of three transgene-free T2 edited lines were significantly different from those of wild-type TBR225. Importantly, one of these $T_2$ lines, harboring the largest homozygous 6-bp deletion, displayed decreased *OsSWEET14* expression as well as a significantly reduced susceptibility to a Vietnamese *Xoo* strains and complete resistance to another one. Our findings indicate that CRISPR/Cas9 editing conferred an improved BLB resistance to a Vietnamese commercial elite rice variety.

**Funding:** This work was supported by the National Technology Innovation Program of Vietnam (Grant No. M.36.DN/18) funded by the Vietnam Ministry of Science and Technology (https://most.gov.vn/vn/pages/Trangchu.aspx) and ThaiBinh Seed Corporation (https://thaibinhseed.com.vn/trang-chu.aspx?lang=en-US). - All equipments, labs and nethouses for this work were supported by Institute of Agricultural Genetics, Vietnam Academy of Agricultural Sciences - Financial support came from the Vietnam Ministry of Science and Technology (https://most.gov.vn/vn/pages/Trangchu.aspx) and ThaiBinh Seed Corporation (https://thaibinhseed.com.vn/trang-chu.aspx?lang=en-US). - The funders had no role in study design, data collection and analysis, decision to publish the manuscript. However, Tran Manh Bao who's employed by ThaiBinh Seed Corporation has contributed to the reviewing and editing the manuscript. His name was added as an author of the manuscript. - The funder (ThaiBinh Seed Corporation) supported the research materials (rice cultivar TBR225) for this study. - The funders provided support in the form of salaries for authors (Pham Xuan Hoi, Nguyen Duy Phuong, Pham Thu Hang and Nguyen Thanh Ha and Tran Manh Bao), but did not have any additional role in the study design, data collection and analysis, decision to publish or preparation of the manuscript. The specific roles of these authors are articulated in the 'author contributions' section.

**Competing interests:** Authors XP, PD, HPT, HNT, and TB are employee of ThaiBinh Seed Corporation. This does not alter our adherence to PLOS ONE policies on sharing data and materials. There are no patents, products in development or marketed products associated with this research to declare.

## Introduction

Bacterial leaf blight (BLB) caused by *Xanthomonas oryzae* pv. *oryzae* (*Xoo*) is a major bacterial disease that causes 10%-20% annual reduction in rice production worldwide [1]. The use of improved rice varieties resistant to *Xoo* is probably the most efficient, economical and environmentally-friendly way to control BLB.

The virulence of *Xoo* depends on the transcriptional activation of specific host disease-susceptibility (*S*) genes by a subgroup of bacterial type III effectors, called transcription activator-like effectors (TALEs) [2]. Upon translocation into the plant cell, TALEs bind to specific host nuclear gene promoter sequences termed Effector-Binding Elements (EBEs) and induce target gene expression to the benefit of the pathogen. The central repetitive domain of TALEs is responsible for DNA target sequence binding. DNA binding involves recognition principles that have been largely deciphered and applied to the computational prediction of TALEs target DNA sequences [3,4]. This and earlier work has fostered the identification of TALEs transcriptional targets in the rice genome and ultimately, of rice BLB *S* genes [2].

All *Xoo* strains recurrently target *S* genes belonging to the *SWEET* gene family and coding for transmembrane sugar exporter proteins [3]. The over accumulation of SWEETs due to TALE induction is presumed to provide an additional ration of apoplastic carbohydrates for full bacterial pathogen multiplication and disease expression [5]. Although all five rice clade III *SWEET* genes can function as *S* genes for bacterial blight, only three, namely *OsSWEET11*, *OsSWEET13* and *OsSWEET14*, are known to be targeted by several unrelated TALEs in nature [6–11]. *OsSWEET11* is activated by PthXo1 [6], *OsSWEET13* is targeted by different variants of PthXo2 [11,12], while *OsSWEET14* is a target of multiple TAL effectors, including AvrXa7, PthXo3, TalC and TalF [7–9].

Previous studies established that rice resistance to *Xoo* resulting from "TALE-unresponsive" alleles can be conferred by natural DNA polymorphisms or targeted editing of EBEs located in *OsSWEET* genes promoters of rice germplasm accessions or engineered rice varieties, respectively [6,13–16]. For example, early resistance engineering work has used TALENs to individually alter the AvrXa7, TalC or TalF EBEs in the *OsSWEET14* promoter and successfully obtained resistance to some Asian *Xoo* strains [13,15]. However, strains collected in Asian countries such as China, Japan, Phillippines, Taiwan, Thailand, India, Nepal or South Korea can express combinations of up to three major TALEs redundantly targeting clade III *OsSWEET* genes with either PthXo3 or AvrXa7 being occasionally associated with PthXo2 [11,17]. Broad BLB resistance engineering thus required multiplex *OsSWEET* promoters EBE editing using the CRISPR/Cas9 system [11,12].

The clustered regularly interspaced short palindromic repeats/CRISPR-associated protein-9 nuclease (CRISPR/Cas9) system is a simple and efficient gene-editing tool developed in the past few years [18,19]. Moreover, the targeted mutations generated by CRISPR/Cas9 can be stably transmitted to the next generation. Thus, CRISPR/Cas9 has become a routine tool in plant laboratories around the world to create various mutants for many applications, including the genetic improvement of crops [20].

BLB is a major rice disease which occurs in many rice cultivating areas of Vietnam [21,22]. Most Vietnamese commercial rice varieties, including TBR225, are susceptible to BLB, resulting in annual yield loss of about 15–30% on average [23]. A few studies have identified rice resistance genes effective against Vietnamese *Xoo* lineages [22,23]. However, no information is currently available on the nature of Vietnamese *Xoo* TALEs and their corresponding *S* genes. Despite the large number of mapped rice BLB resistance genes [24,25], there is a need for alternative breeding approaches that enable the rapid introduction of broad BLB resistance in elite varieties in order to cope with swift pathogen populations adaptive shifts in the fields [11,26].

Here, we report on the identification of *OsSWEET14* as a transcriptional target of Vietnamese *Xoo*. CRISPR/Cas9-mediated mutagenesis of the *OsSWEET14* promoter in TBR225, a major elite variety in rice production areas of North Vietnam, is shown to confer BLB resistance without detectable yield penalty. The current study found this quintessential *S* gene to be associated with the virulence of Vietnamese *Xoo* strains. This is an important step for the future design and implementation of broad-spectrum BLB-resistance in elite rice varieties using genome editing in Vietnam.

## Materials and methods

### Plant and pathogen materials

Rice cultivar TBR225 (*Oryza sativa* L. ssp. *indica*) were obtained from ThaiBinh Seed Cor. [27]. All edited and wild-type (WT) TBR225 plants were grown in a net-house under the following average conditions: 30˚C for 14 h (light) and 25˚C for 10 h (dark) with 80% humidity. The *Xoo* VXO_11 and VXO_15 strains used in this study were isolated from diseased leaves collected in Hanoi-Vietnam in 2013 and 2016, respectively. Bacteria were cultured as described in Zhou et al. (2015) [28].

### Gene expression analysis

Gene expression analyses were carried out as described previously [29] by RT-PCR method. The rice leaves were infiltrated with the indicated bacterial strains and used for total RNA extraction 48 h post inoculation using the TRIzol reagent (Invitrogen, USA). One microgram of RNA was used for each RT-PCR with oligo (dT) primer followed by PCR with *OsSWEET14*-specific primers (forward 5′-ACTTGCAAGCAAGAACAGTAGT-3′ and reverse 5′-ATGTTGCCTAGGAGACCAAAGG-3′). An *Eppendorf Mastercycler ep Gradient S* was used for 35 PCR cycles. The *OsEF1α* gene was used as a constitutive control [15] using specific primers (forward 5′-GAAGTCTCATCCTACCTGAAGAAG-3′ and reverse 5′-GTCAAGAGCCTCAAGCAAGG-3′).

### gRNA design

The *OsSWEET14* promoter (GenBank, accession number: AP014967.1) was amplified by PCR with forward primer 5'-TTGCGGCTCATCAGTTTCTC-3' and reverse primer 5'-CTAGGAGACCAAAGGCGAAG-3' from genomic DNA of TBR225 rice plants and ligated in pGEM-T Easy vector (Promega) for sequencing. The gRNA target sequence (Fig 1A) for editing the TBR225 *OsSWEET14* promoter was designed based on the sequence of the cloned TBR225 *OsSWEET14* promoter using a combination of two bioinformatics tools CRISPR-P v2.0 [30] and CCTop [31]. A gRNA sequence with high on-target and low off-target scores in both prediction tools was chosen for vector construction.

### Vector construction

The Cas9 rice expression vector (pUbi-Cas9) [32] and the sgRNA expression vector (pENTR-sgRNA) under the control of the *OsU6* promoter [33] were used to construct the pCas9/OsSWEET14-gRNA expression vector. The complementary oligonucleotides with appropriate 4-bp overhangs were synthesized by Macrogen (Korea). After heat denaturation, the complementary oligonucleotides (5′-gtgtGGTGCTAAGCTCATCAAGCC-3′ and 5′-aaacGGCTTGATGAGCTTAGCACC-3′) were first annealed to each other, phosphorylated, and ligated into the *Bsa*I-digested vector pENTR-sgRNA. The integrity of the inserted fragment was verified by sequencing. Subsequently, the sgRNA cassette was cloned into pUbi-Cas9 using the Gateway

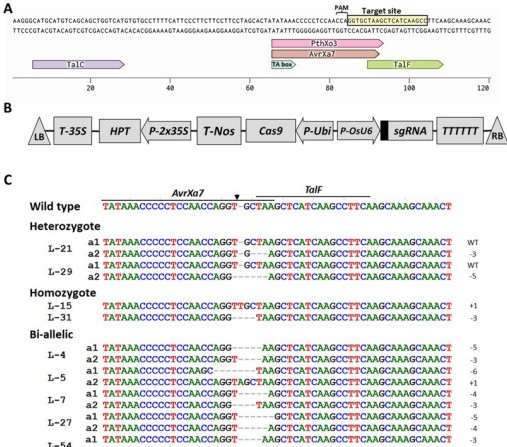

**Fig 1. CRISPR/Cas9-induced *OsSWEET14* promoter modification in TBR225 rice.** (A) A region of the *OsSWEET14* promoter containing four EBEs (TalC, PthXo3, AvrXa7 and TalF) and putative TATA box from TBR225. The target site (complementary to the guide RNA) is shown in the box, immediately following the protospacer adjacent motif (PAM). (B) T-DNA region of the CRISPR/Cas9-mediated genome editing construct carrying OsSWEET14-sgRNA (indicated by the black box). The expression of *Cas9* is driven by the maize *ubiquitin* promoter (P-Ubi); the expression of the OsSWEET14-sgRNA is driven by the rice *OsU6* promoter (P-OsU6a); the expression of *HPT* is driven by two *CaMV35S* promoters (P-2×35S); T-35S, T-Nos and TTTTTT: Gene terminators; LB and RB: Left and right border, respectively. (C) Alignment of the *OsSWEET14* promoter fragment in the nine $T_0$ transgenic TBR225 rice plants edited in the AvrXa7, PthXo3 and TalF EBEs. The lines on top of the wild-type sequence represent the binding sites of AvrXa7, PthXo3 and TalF. The arrow indicates the expected cutting site of the Cas9 complex used in this study. The labels on the left indicate the name of examined mutant lines; (a1) and (a2) distinguish alleles in the same line. The numbers on the right indicate the type of mutation and the number of nucleotides involved; (+) and (-) indicate insertion and deletion, respectively.

LR clonase (Life Technologies) (Fig 1B). The resulting construct was confirmed by Sanger sequencing of the insertion junctions.

## *Agrobacterium*-mediated rice transformation

The pCas9-OsSWEET14-gRNA was electroporated into *Agrobacterium tumefaciens* EHA105 and the resulting strain was used to transform rice using the method described by Hiei et al. (1994) [34]. The presence of the transgene in the genome of $T_0$ hygromycin-resistant plants or segregating $T_1$ individuals was evaluated by PCR using 5′-ATGGCCCCAAAGAAGAAG-3′ and 5′- GCCTCGGCTGTCTCGCCA-3′ primers specific for *Cas9*. T1 individuals were analyzed by PCR using *Cas9*, *OsSWEET14-gRNA* (5′- GGATCATGAACCAACG-3′ and 5′- GAATTCGATAT CAAGCTT-3′) and *HPT* (5'-AAACTGTGATGGACGACACCGT-3' and 5′- GTGGCGATCCTGC AAGCTCC -3′) specific diagnostic primer pairs together with a positive control pair (5′-TTG CGGCTCATCAGTTTCTC-3′ and 5′- TGGATCAGATCAAAGGCAAC -3′) specific to the *OsS-WEET14* promoter.

## Bacterial blight inoculation

Rice cultivation and disease assays were done according to the methods of Blanvillain-Bau-fumé et al. (2017) [15]. Bacteria were cultured in PSA media (10 g/liter peptone, 10 g/liter sucrose, 1 g/liter glutamic acid, 15 g/liter Bacto Agar) at 28°C for two days [35] and inoculated at an optical density ($OD_{600}$) of 0.5 (infiltrations) or 0.4 (leaf clipping) in water. For lesion length measurements, at least three inoculated leaves per plant and three plants for each line were measured 14 days after inoculation (DAI), and scored as follows: high resistance (lesion length < 8 cm), moderate resistance (lesion length 8–12 cm) and susceptibility (lesion

length > 12 cm). For gene expression analyses, 4-cm leaf sections infiltrated with bacterial suspensions were collected at 48 h after inoculation for RNA extraction. Experiments included samples from three pooled biological replicate leaves. The plants inoculated with distilled water only were used as negative controls.

### Analysis of *OsSWEET14* edited allele sequences

To determine the nature of the mutation at the target site, all transgenic $T_0$ or $T_1$ plants were analyzed by PCR using genomic DNA (50 ng) as a template and *OsSWEET14* specific primers (5′-TTGCGGCTCATCAGTTTCTC-3′ and 5′- TGGATCAGATCAAAGGCAAC -3′). The PCR products were directly sequenced using the Sanger method. The sequencing chromatograms were decoded using the Degenerate Sequence Decoding method [36] in order to identify the mutations.

### Evaluation of major agronomic traits under net-house conditions

WT and selected mutant plants were planted under net-house conditions in a randomized pot design experiment. At maturity, five plants of each line were investigated for the following agronomic traits: growth duration, plant height, number of tillers per plant, number of grains per panicle, number of filled grains per panicle and yield (seed mass) per plant. The experiment was repeated three times, so a total of fifteen plants were evaluated for each line.

### Analysis of potential off-target editing

Off-target sequences were predicted with the CCTop tool [31] against the *OsSWEET14* promoter sgRNA and the rice Nipponbare genome with default parameters. A total of 18 potential off-target sequences were identified. Three of them were located in coding regions (S2 Table). These regions were amplified by PCR using the specific primers listed in S2 Table and analyzed by sequencing.

## Results

### Vietnamese *Xoo* strains induce *OsSWEET14* during infection of the TBR225 rice variety

*OsSWEET14/Os11N3* was previously identified as a susceptibility gene for *Xoo* strains relying on either of the AvrXa7, PthXo3, TalF (formerly Tal5) or TalC TALEs for infection of the rice cultivars Nipponbare and Kitaake [11]. Because *Xoo* strains tend to frequently target this gene, we first sequenced a region of the *OsSWEET14* promoter from rice cultivar TBR225 to examine if it also carries documented target EBEs. Based on the Nipponbare genome sequence in database (AP014967.1), the region encompassing 1343 bp sequence upstream and 52 bp sequence downstream of the predicted transcription start site of *OsSWEET14* gene from TRB225 rice cultivar was PCR amplified and sequenced (S1 Fig). The promoter region including the putative TATA box (TATAAA) and the AvrXa7, PthXo3, TalF/Tal5 and TalC EBEs (Fig 1A), located 319 bp to 216 bp upstream of the ATG initiation codon, showed 100% identity to the Nipponbare sequence. This therefore implied that in principle, the TBR225 *OsSWEET14* promoter can be recognized by characterized major *Xoo* TALEs.

As shown in Fig 2A, we also challenged TBR225 plants with two Vietnamese *Xoo* strains VXO_11 and VXO_15, both originating from the Hanoi province, using leaf clipping assays. We consistently obtained typical extended disease lesions 14 days after inoculation (25.5 cm and 26.6 cm average lesions length for VXO_11 and VXO_15, respectively in the experiment of S2 Fig), indicating that the TBR225 variety is susceptible to BLB.

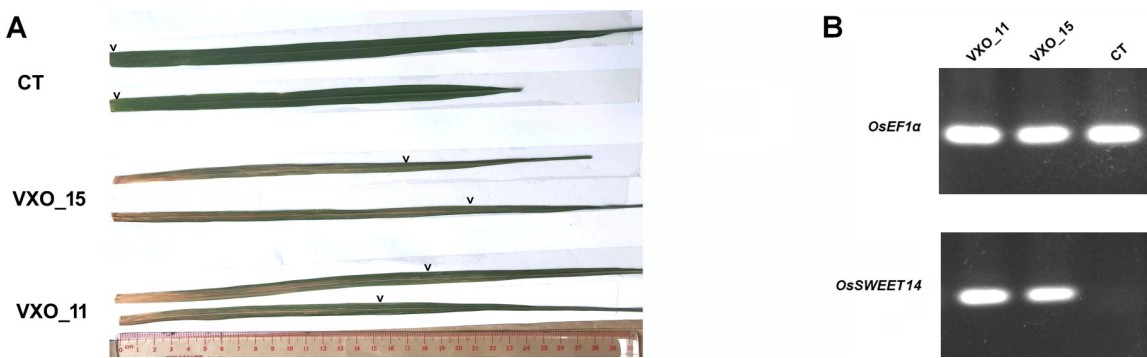

**Fig 2. *OsSWEET14* is likely a susceptibility gene for Vietnamese *Xoo* strains in rice cultivar TBR225.** (A) Representative images of the disease lesions obtained 14 days after leaf clipping inoculation of TBR225 rice leaves with Vietnamese *Xoo* strains VXO_11 and VXO_15 or with water (CT). The chevrons above the leaves indicate the maximum visible extent of lesions away from the inoculation point on the left (B). *OsSWEET14* expression pattern obtained by RT-PCR two day post-infiltration of TBR225 rice leaves with Vietnamese *Xoo* strains. CT Plants were inoculated with water only. The experiment was repeated three times.

To test if *OsSWEET14* is a potential direct virulence target of Vietnamese *Xoo* strains, we infiltrated TBR225 rice leaves with the two Vietnamese *Xoo* strains. Forty-eight hours post infiltration, TBR225 plants inoculated with VXO strains displayed a strong induction of *OsSWEET14* relative to water controls (Fig 2B). These results suggest that *OsSWEET14* is a transcriptional target of VXO strains and that it may act as a susceptibility gene in TBR225.

## CRISPR/Cas9 design for *OsSWEET14* promoter editing

Our main objective was to engineer resistance to BLB caused by Vietnamese *Xoo* strains. To this end, we subsequently sought to specifically modify the *OsSWEET14* promoter in TBR225 rice with CRISPR/Cas9-mediated editing. Previous work revealed that while African *Xoo* strains rely on TalC and occasionally, TalF, all Asian *Xoo* strains use either PthXo3- or Avr-Xa7-like TALEs to activate *OsSWEET14* [11]. Because the *talC* gene is currently exclusively found in African strains, we reasoned that it is unlikely that Vietnamese strains carry a *talC* copy. Thus, to maximize our chances to perturb all remaining documented EBEs, we selected a 20-bp nucleotide target site overlapping the PthXo3, AvrXa7 and TalF EBEs and having a predicted cut site located near the 3'-end of the AvrXa7 EBE (Fig 1A). The recombinant binary plasmid pCas9/OsSWEET14-gRNA for CRISPR/Cas9 mediated editing of *OsSWEET14* was transformed into the rice variety TBR225 via *Agrobacterium*-mediated transformation (S1 Table). A total of nine TBR225 transformants were selected from 10 independent PCR-validated transgenic $T_0$ TBR225 plants to further investigate CRISPR/Cas9-targeted mutagenesis of the *OsSWEET14* promoter. In order to decipher the nature of the editing events in *OsSWEET14*, the promoter sequencing data of transgenic lines were analyzed using the Degenerate Sequence Decoding software [36]. All 9 $T_0$ transgenic plants harbored at least an editing event (Fig 1C): two were heterozygous mutant/wild type, two had homozygous mutations, and five had bi-allelic mutations. Regarding the type of mutations, 66.7% were nucleotide deletions, 11.1% of the mutations were nucleotide insertions and no substitution was detected (Table 1).

## Inheritance of CRISPR/Cas9-induced mutations in the $T_1$ generation

To assess the inheritance of the CRISPR/Cas9-induced *OsSWEET14* mutations in the next generation, all $T_0$ mutant transgenic plants (Fig 1C) were allowed to self-pollinate, and $T_1$ transgenic plants were randomly selected in the progeny of $T_0$ plants for sequencing and

**Table 1. Frequencies of mutant genotypes and target mutation types in $T_0$ transgenic plants.**

| Mutant genotype ratios[a] (%) | | | Mutation type ratios[b] (%) | | |
|---|---|---|---|---|---|
| **Heterozygote** | **Homozygote** | **Bi-allelic** | **Deletion** | **Insertion** | **Substitution** |
| 22.2 (2/9) | 22.2 (2/9) | 55.6 (5/9) | 66.7 (12/18) | 11.1 (2/18) | 0 (0/18) |

[a] (Number of on-target mutant genotype/total number of on-target mutant genotypes) x 100%.

[b] (Number of allele mutation type/number of all allele mutation types) x 100%.

analysis of their edited site (Table 2). All $T_1$ individuals derived from $T_0$ plants previously genotyped as homozygous possessed the same allele as their parent, suggesting stable inheritance of the mutations to the next generation. Similarly, the T1 progeny of each of both bi-allelic and heterozygous mutation $T_0$ lines showed a segregation ratio which is consistent with Mendelian segregation ($\chi^2 < \chi^2_{0.05, 2} = 5.99$), indicating that the CRISPR/Cas9-induced mutations in $T_0$ plants were transmitted as expected to the next generation. Interestingly, no new mutant allele was detected in the $T_1$ generation of both heterozygous mutants L-21 and L-27, even though most of them still carried the transgene. Overall, consistent with previous similar studies, our results indicate that the CRISPR/Cas9-mediated mutations generated here are stably transmitted to the next generation in a Medelian fashion.

### Selection of transgene-free mutant TBR225 rice lines

To identify T-DNA free $T_1$ rice plants containing a mutation in EBEs of the *OsSWEET14* promoter, PCR analysis was carried out using primers specific to *Cas9*, sgRNA and *HPT* sequences (Table 2). A T1 individual was considered devoid of the transgene if the control amplification of the *OsSWEET14* promoter was successful and if none of the PCR reactions with independent primer pairs designed on the T-DNA produced a detectable diagnostic band. The results of this PCR screen show that the T-DNA could be segregated out in the progeny of most $T_0$ lines, with 88.9% of the $T_0$ lines generating T-DNA-free progeny. In total, 44 of 221 analyzed edited $T_1$ plants did not generate a specific amplicon from the T-DNA construct and 15 of them were homozygous mutant harboring the desired *OsSWEET14* modifications. Our results demonstrate that transgene-free, homozygous mutant individuals could be obtained in the segregating progeny of selfed $T_0$ individuals.

**Table 2. Transmission of CRISPR/Cas9 editing events to the $T_1$ generation.**

| $T_0$ plant | Genotype | Allele(s) | No. of $T_1$ plants tested | Mutation inheritance in the $T_1$ generation | | No. of T-DNA-free plants |
|---|---|---|---|---|---|---|
| | | | | **Alleles segregation** | **$\chi^2$ (1:2:1)** | |
| L-4 | Bi-allelic | -5/-3 | 32 | 10 (-5), 18 (-5/-3), 4 (-3) | 2,750 | 5 (2*) |
| L-5 | Bi-allelic | -6/+1 | 44 | 9 (-6), 22 (-6/+1), 13 (+1) | 0,727 | 10 (2*) |
| L-7 | Bi-allelic | -4/-3 | 38 | 14 (-4), 17 (-4/-3), 7 (-3) | 3,000 | 11 (4*) |
| L-15 | Homozygote | +1 | 5 | 5 (+1) | - | 1 (1*) |
| L-21 | Heterozygote | -3 | 26 | 3 (-3), 13 (-3/wt), 10 (wt) | 3,769 | 7 (1*) |
| L-27 | Bi-allelic | -5/-4 | 7 | 1 (-5), 3(-5/-4), 3 (-4) | 1,286 | 0 |
| L-29 | Heterozygote | -5 | 33 | 6 (-5), 19 (-5/wt), 8 (wt) | 1,000 | 2 (0*) |
| L-31 | Homozygote | -3 | 15 | 15 (-3) | - | 5 (5*) |
| L-54 | Bi-allelic | -3/-2 | 21 | 3 (-3), 12 (-3/-2), 6(-2) | 1,286 | 3 (0*) |

"+" and "-" indicate respectively, insertion and deletion, of the indicated number of nucleotides.

"w", wild type.

*Number of homozygous mutant plants without T-DNA.

## TBR225 *OsSWEET14* promoter editing confers resistance to Vietnamese *Xoo*

To characterize the BLB-resistance phenotype of the generated rice mutants, three T-DNA-free, homozygous TBR225 edited lines, namely, L-5.7(-6), L-31.12(-3) and L-15.4(+1) with *OsSWEET14* promoter alleles corresponding respectively to L-5-a1 (6bp deletion), L-31 (3bp deletion) and L-15 (1bp insertion) in Fig 1C, were established. Selected $T_1$ individuals were propagated to obtain $T_2$ seeds which were used to perform BLB susceptibility assays. Edited $T_2$ and WT TBR225 plants were inoculated by leaf-clipping with the VXO_11 and VXO_15 strains at the eight-week stage. The inoculated leaves of wild type TBR225 plants and of edited lines L-15.4(+1) and L-31.12(-3) developed long water-soaked lesions typical of BLB, ranging from 18.3 cm to 29.0 cm in length. In contrast, the edited line L-5.7(-6), harboring a longer 6-bp deletion at the target site, displayed high (1.2 cm average lesion length) and moderate (7.3 cm average lesion length) resistance to VXO_11 and VXO_15 strains, respectively (Fig 3). Means comparisons with a Tukey's HSD test further indicated that irrespective of the inoculated strain, the mean lesion lengths measured on the L-15.4 (+1), L-31.12(-3) or wild type lines were not significantly different. In contrast, the mean lesion lengths recorded on the L-5.7(-6) mutant line were significantly different from those obtained on the wild type and the two other edited lines challenged with either of the Vietnamese strains (Fig 3B). Furthermore, our off-target editing analysis on line L-5.7(-6) did not reveal unintended modifications of other annotated rice loci (S2 Table and S5 Fig), indicating that the 6-bp deletion in the *OsSWEET14* promoter is probably responsible for this phenotype.

Consistent with disease assays and as shown in Fig 3C, whereas a semiquantitative RT-PCR signal for *OsSWEET14* expression was detected on the parental variety and the L-15.4(+1) and L-31.12(-3) edited lines following VXO_11 and VXO_15 infiltration, this amplicon was undetectable in the resistant L-5.7(-6) line.

In conclusion, this data shows that the 6-bp deletion in the AvrXa7/PthXo3 EBE reduces dramatically *OsSWEET14* expression following VXO strains inoculation and confers resistance to these strains. In contrast, shorter modifications on the 3'-end of this EBE are insufficient to perturb *OsSWEET14* expression after inoculation and do not confer detectable protection against the corresponding strains. Finally, while these results strongly support the view that *OsSWEET14* functions as a unique susceptibility gene in the interaction between strain VXO_11 and the TBR225 rice variety, the resistance to strain VXO_15 is not as dramatic and may suggest that other mechanisms partially counteract the effects of the AvrXa7/PthXo3 EBE 6-bp deletion in edited TBR225 plants.

## TBR225 *OsSWEET14* promoter edited lines agronomic performances are undistinguishable from the parental variety

To determine if mutations in the *OsSWEET14* promoter affect agronomic traits of TRB225 rice plants, three independent homozygous mutant lines were analyzed by measuring their growth duration, plant height, number of tillers per plant, number of grains per panicle, number of filled grains per panicle, yield per plant and amylose content under net-house conditions (see picture of S3 Fig). ANOVA tests and Student's *t* tests showed that the mutant lines displayed no significant difference to TBR225, in terms of the examined agronomic traits, under our net-house conditions (Table 3). These results suggest that the tested CRISPR/Cas9-induced mutations in the *OsSWEET14* promoter did not negatively impact the main agronomic traits of TBR225.

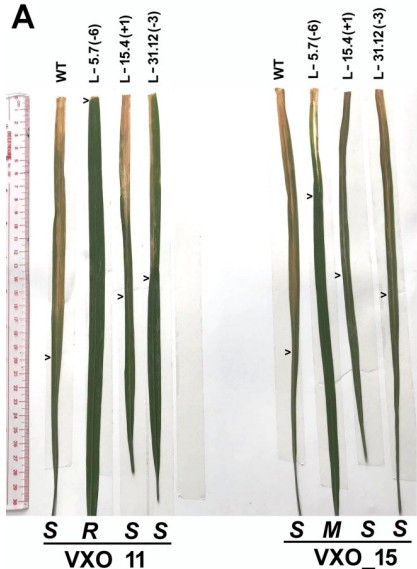
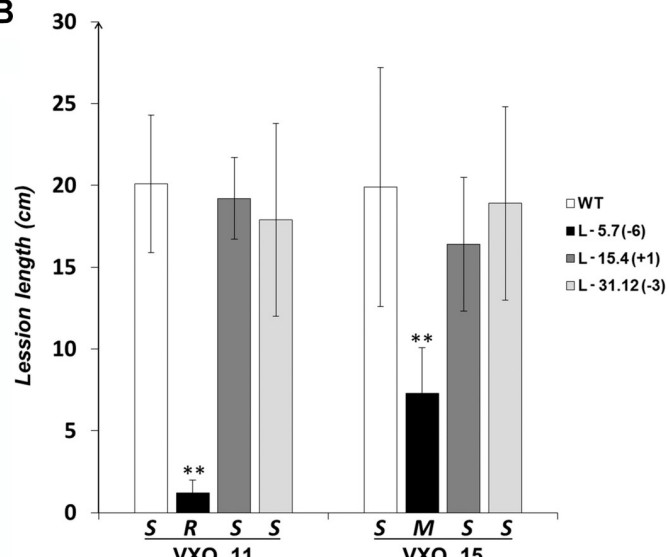

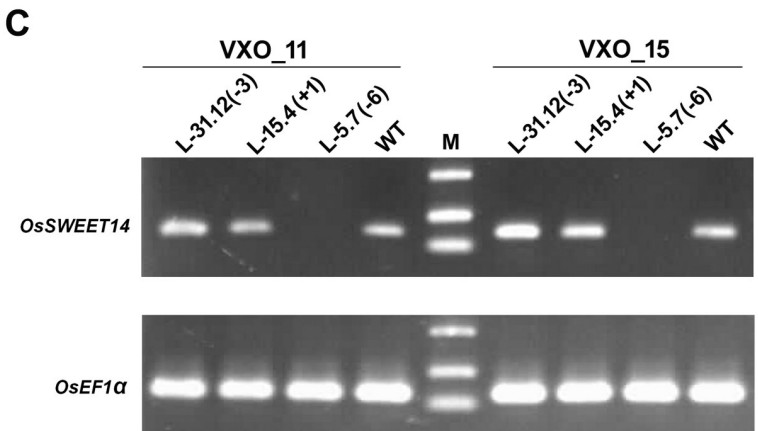

**Fig 3. BLB resistance assays for homozygous mutant rice lines L-5.7(-6), L-15.4(+1) and L-31.12(-3).** (A) Leaves were photographed 14 days post-leaf clipping inoculation of *Xoo* strains VXO_11 and VXO_15; arrow heads indicate the end of the lesion. (B) Mean lesion lengths (bars) and standard deviations (error bars). Values were measured 14 days post-leaf clipping inoculation of two *Xoo* strains VXO_11 and VXO_15 and were computed from at least three leaves from each of three plants. Asterisks indicate significant differences relative to wild type plants (Tukey's HSD test; $^{**}P < 0.05$). The number in the parentheses following the line name indicates the type of mutation and the number of nucleotides involved. The letters above strain labels indicate susceptibility score (R—high resistance; M–moderate resistance; S—susceptibility). The experiment was repeated three times. (C) *OsSWEET14* expression pattern obtained by RT-PCR two day post-infiltration of genome edited homozygous mutant rice lines L-31.12(-3), L-15.4(+1) and L-5.7(-6) and parental TBR225 rice leaves with Vietnamese *Xoo* strains. This experiment was repeated two times with similar results.

## Discussion

Recently, the CRISPR/Cas9 system has emerged as a powerful tool for gene editing in many organisms including plants. Because of its specificity and efficiency, this system has been widely used to improve important agronomic traits of major crops such as rape, tomato, soybean, rice, wheat and maize [37]. Excluding easy-to-transform reference accessions such as Nipponbare and Kitaake that are widely used in the laboratory, the number of reports on the

**Table 3. Agronomic traits evaluation of homozygous T$_2$ mutant lines.**

| Lines | Growth duration (day) | Plant height (cm) | No. of tillers per plant | No. of grains per panicle | No. of filled grains per panicle | Amylose content (%) |
|---|---|---|---|---|---|---|
| WT | 108.4 ± 1.1[a] | 86.6 ± 3.2[a] | 5 ± 0.7[a] | 144.4 ± 4.9[a] | 125 ± 4.5[a] | 13.2 ± 0.38[a] |
| L-5.7(-6) | 108 ± 1.2[a] | 86.4 ± 4.3[a] | 5.2 ± 0.4[a] | 144.2 ± 4.4[a] | 123.4 ± 5.5[a] | 13.7 ± 0.35[a] |
| L-15.4(+1) | 107.8 ± 0.8[a] | 86.4 ± 5.0[a] | 4.8 ± 0.4[a] | 147.8 ± 5.1[a] | 121.8 ± 3.0[a] | 13.5 ± 0.41[a] |
| L-31.12 (-3) | 108 ± 1.2[a] | 88.4 ± 4.3[a] | 5.4 ± 0.5[a] | 144.6 ± 5.3[a] | 124.2 ± 7.4[a] | 13.8 ± 0.21[a] |

Five plants per line were measured. Experiments were repeated three time.

Means followed by the same letter do not differ significantly ($P < 0.05$).

improvement of agriculturally relevant elite rice cultivars for pertinent traits using the CRISPR/Cas9 technology (see for example [38–42]) is gradually increasing but is still limited.

TBR225 [27], a major commercial rice variety cultivated in large areas of Northern Vietnam, has the advantages of early maturity, high and stable yield, as well as cooking quality. However, it is very susceptible to BLB. Here, the CRISPR/Cas9-mediated editing method was applied in order to rapidly improve the BLB resistance of TBR225 by modifying the AvrXa7, PthXo3 and TalF EBEs on the promoter of *OsSWEET14*. Of the three generated homozygous mutant lines tested for resistance, the one carrying the largest deletion at the target site (6 bp) showed a significantly improved resistance to infection with two *Xoo* strains VXO_11 and VXO_15. Therefore, using the major commercial rice variety TBR225 as an example, we illustrate the advantages of CRISPR/Cas9 tool for rice breeding.

In the present study, the frequency of individuals with CRISPR/Cas9-induced mutations in T$_0$ transgenic plants was 90%, which is similar to previous observation [33]. We obtained only two heterozygous mutant/wild type lines versus seven homozygous or bi-allelic mutant lines. This high frequency of mutated alleles is another proof that the CRISPR/Cas9 system is indeed an efficient tool for gene editing in plant. We also observed the stable transmission of edited alleles to subsequent generations. This is a common phenomenon that has been repeatedly documented for rice plants carrying CRISPR/Cas9-induced mutations [38,40,43,44]. In this study, we obtained only two types of induced mutations in T$_0$ plants: insertion (11.1%) and deletion (66.7%), but no substitution were observed. In some earlier studies, new mutations were continuously obtained in the T$_1$ offspring of heterozygous T$_0$ mutants because the Cas9 complex remains active on edited targets until the seed or PAM regions cease to be functional [35,37,43]. In contrast, here, all the T$_1$ plants generated from both heterozygous lines L-21 and L-29, regardless of whether they had a CRISPR/Cas9 T-DNA transgene integrated in their genome, did not show any new mutation. We could also readily obtain transgene-free plants from most of the T$_1$ segregation populations without any laborious crossing or backcrossing steps, which illustrates an advantage of the CRISPR/Cas9 technology compared to conventional breeding.

Clade III SWEET family proteins are involved in a number of biological processes such as seed and pollen development or pathogen susceptibility [45]. Their inactivation has previously been shown to cause pleiotropic and/or detrimental effects. For example, both *ossweet11* single and *ossweet11-ossweet15* double Kitaake rice mutants showed defects in endosperm development and filling [46]. In addition, RNA-mediated silencing of either *Os11N3/OsSWEET14* [7] or *Os8N3/OsSWEET11* [6] in BLB resistant Kitaake lines causes negative effects on seed production. In contrast, here, we show that T-DNA-free TBR225 plants harboring homozygous mutations generated with the CRISPR/Cas9 system in the AvrXa7/PthXo3 EBE of the *OsSWEET14* promoter exhibited enhanced *Xoo* resistance but did not show any significant

difference in all examined agronomic traits compared to wild-type plants under net-house growth conditions. It is conceivable that limited modifications in promoter regions do not affect the normal expression of *SWEET* genes in contrast to KO or silenced lines. Our findings are consistent with the previous work of Oliva et al. [11] who studied 30 combinations of EBE mutations in the *OsSWEET11*, *OsSWEET13* and *OsSWEET14* promoters of the IR64 or Ciherang-Sub1 varieties and detected only a single line with abnormal agronomic traits.

Some individual *Xoo* strains have evolved a set of distinct TALE effectors that collectively target several members of the clade III SWEET family. The presence of these redundant TALES thereby trumps single "loss-of-tale-responsiveness" resistance alleles [11,12,17,47]. For example, Kitaake lines carrying TALEN-induced mutation in the *SWEET14* promoter [13,15] exhibit resistance to strains which depend exclusively on matching AvrXa7/PthXo3 for clade III *SWEET* family induction. Likewise, the natural *xa13* allele [48] or CRISPR/Cas9-induced mutation in the *SWEET11* promoter [11] exhibit resistance to strains such as PXO99 which depend exclusively on PthXo1, for virulence. However, the BLB resistance of the Kitaake lines harboring mutations in both AvrXa7/PthXo3 (*OsSWEET14*) and PthXo1 (*OsSWEET11*) EBEs was defeated by *Xoo* strains expressing simultaneously the AvrXa7/PthXo3 and PthXo2B TALEs [11]. Recently, the stacking of EBE-edited alleles in several *OsSWEET* promoters have overcome this limitation and was shown to achieve a broad spectrum of resistance to strains from most BLB-prone countries in Asia [11,12].

All of the three $T_2$ lines tested for BLB resistance were affected for the AvrXa7/PthXo3 EBE and conserved an otherwise wild type TalF EBE (Fig 1C). The homozygous mutant TBR225 line L-5.7(-6) carrying a 6-bp deletion in the AvrXa7/PthXo3 EBE exhibited a significantly enhanced resistance to two Vietnamese *Xoo* strains compared to WT TBR225. The L-15.4(+1) and L-31.12(-3) lines that harbored more subtle alterations in the 3'-end of this EBE (a 1-bp insertion and a 3-bp deletion, respectively) in contrast remained susceptible to VXO strains. Our *OsSWEET14* expression analysis after Vietnamese Xoo strains inoculation (Fig 1C) suggests that these editing events did not alter the EBE sequence sufficiently to compromise promoter recognition by an AvrXa7/PthXo3-like Vietnamese TALE. With less than 2 cm average lesion length, the resistance of line L-5.7(-6) (6-bp deletion) to the VXO_11 strain is rather extreme (versus average lesion length of 20.1 cm on wild type plants). Moreover, in this line, *OsSWEET14* expression following bacterial inoculation is strongly reduced relative the parental line and the two other edited lines, which suggest that in this case, recognition by an AvrXa7/PthXo3-like Vietnamese TALE is abrogated. Consistent with *OsSWEET14* expression analysis and as shown in S4 Fig, the Talvez [49] target prediction scores for AvrXa7 and PthXo3 on the *OsSWEET14* promoter L-5-a1 allele sequence of line L-5.7(-6) are markedly lower than on the wild type promoter sequence. This is not the case however for the edited alleles carried by lines L-15.4(+1) and L-31.12(-3) (respectively L-15 and L-31 in S4 Fig) whose Talvez scores are identical or slightly lower than those of the wild type promoter sequence.

The magnitude of the effect of the 6-bp deletion allele on susceptibility to VXO_11 is comparable to the dramatic effect of previously characterized alterations of the same EBEs in the Kitaake background against the PXO86 strain that possesses a single TALE, AvrXa7, targeting *OsSWEET14* for clade III *OsSWEET* gene induction [15]. By analogy, this suggests that *OsSWEET14* is also the only clade III *OsSWEETs* target of VXO_11 in the TBR225 background but, in order to confirm this hypothesis an examination of other clade III *OsSWEET* genes expression patterns in response to this strain would be required. The situation with the VXO_15 strain is not as straightforward to interpret and will require further investigations. Although the 6-bp deletion in the AvrXa7/PthXo3 EBE did provide an increased resistance to the edited plants, the VXO_15 strain caused intermediate disease severity (7.3 cm average lesion length on Fig 3). This incomplete resistance is unlikely to result from the partial but still

productive recognition of subsequences of the altered EBE by a VXO_15 AvrXa7/PthXo3-like TALE because *OsSWEET14* expression is similarly decreased in response to either this strain or VXO_11 (Fig 3C). Alternatively, contrary to all Asian *Xoo* examined so far, but similar to African *Xoo* [11,15], VXO_15 may have the intrinsic potential to cause disease in the absence of clade III *OsSWEET* gene induction, a phenomenon that seems to be dependent on the edited rice variety genetic background [42]. More likely, analogous to other Asian strains, VXO_15 may encode alternative TALES, such as PthXo2B or PthXo1 that compensate the loss of *OsSWEET14* induction by targeting other clade III *OsSWEET* genes. In this regard, deciphering clade III *OsSWEET* genes expression patterns in combination with long read genome sequencing will ultimately help describe TALEs variability in Vietnamese *Xoo* strains and its functional impact on *OsSWEET* genes induction.

In conclusion, we showed that editing specific EBEs of *Xoo* TALEs via CRISPR/Cas9 tool is an efficient method for improving BLB resistance of elite rice varieties such as TBR225 without detectable yield penalties. This also uncovered the potential diversity of TALEs in Vietnamese *Xoo* population, which will thus require future investigations to address the TALE repertoires of Vietnamese *Xoo* strains in order to generate broad-spectrum BLB-resistant rice varieties in Vietnam.

## Supporting information

**S1 Fig. Nucleotide sequence of the *OsSEET14* promoter in TBR225.**
(TIF)

**S2 Fig. Virulence of Vietnamese *Xoo* strains VXO_11 and VXO_15 on TBR225 rice.** Grey points correspond to individual lesion length measurements while the black points indicate the calculated average value. The line range represents standard deviation.
(TIF)

**S3 Fig. Picture of an individual plant from the homozygous mutant rice lines L-5.7(-6).**
(TIF)

**S4 Fig. Talvez scoring of AvrXa7, PthXo3 and TalF target EBES in the edited *OsSWEET14* promoter allele sequences.** Score values are represented both by the length of the horizontal bar and a fill color scale. Higher Talvez prediction scores reflect a better match between a predicted EBE and the sequence of RVD of the query TALE.
(TIF)

**S5 Fig. Amplicon sequencing of predicted off-target sites for the *OsSWEET14* promoter-sgRNA in annotated exons of the TBR225 edited line L-5.7(-6).** Potential unintended target sequences including the PAM are highlighted in boxes. They are all identical to the expected wild type Nipponbare sequences.
(TIF)

**S1 Table. Key figures on the TBR225 transformation procedure for *OsSWEET14* promoter editing.**
(DOCX)

**S2 Table. Output of the CCTop tool used with the *OsSWEET14* promoter sgRNA for off-target prediction on the rice Nipponbare genome.**
(DOCX)

**S1 Raw images. Original photograph used in Fig 2 for the RT-PCR gels panel.**
(DOCX)

**S2 Raw images. Original photograph used in Fig 3C for the RT-PCR gels panel.**
(DOCX)

## Acknowledgments

We are grateful to Msc. Pham Thi Van, Dr. Cao Le Quyen and Dr. Nguyen Van Cuu from the Institute of Agricultural Genetics for rice transformation experiments, Msc. Nguyen Thi Thu Ha from the Institute of Agricultural Genetics for managing the *Xoo* strains collection and Msc. Nguyen Thi Nhung from Thaibinh Seeds Cor. for kindly providing the rice accessions.

## Author Contributions

**Conceptualization:** Phuong Nguyen Duy, Sebastien Cunnac, Xuan Hoi Pham.

**Data curation:** Phuong Nguyen Duy.

**Formal analysis:** Phuong Nguyen Duy, Dai Tran Lan, Sebastien Cunnac, Xuan Hoi Pham.

**Funding acquisition:** Phuong Nguyen Duy, Tran Bao Manh, Xuan Hoi Pham.

**Investigation:** Phuong Nguyen Duy, Dai Tran Lan, Hang Pham Thu, Huong Phung Thi Thu, Ha Nguyen Thanh, Ngoc Phuong Pham, Florence Auguy, Sebastien Cunnac.

**Methodology:** Phuong Nguyen Duy, Dai Tran Lan, Hang Pham Thu, Ha Nguyen Thanh, Sebastien Cunnac, Xuan Hoi Pham.

**Project administration:** Xuan Hoi Pham.

**Resources:** Hang Pham Thu, Ha Nguyen Thanh, Florence Auguy.

**Supervision:** Sebastien Cunnac, Xuan Hoi Pham.

**Validation:** Sebastien Cunnac, Xuan Hoi Pham.

**Writing – original draft:** Phuong Nguyen Duy, Dai Tran Lan.

**Writing – review & editing:** Huong Bui Thi Thu, Tran Bao Manh, Sebastien Cunnac, Xuan Hoi Pham.

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
