## [Decision Letter · Decision Letter 0]

13 Nov 2020

PONE-D-20-33201

Improved bacterial leaf blight disease resistance in the major elite Vietnamese rice cultivar TBR225 via editing of the OsSWEET14 promoter

PLOS ONE

Dear Dr.Hoi Xuan Pham,

Thank you for submitting your manuscript to PLOS ONE. After careful consideration, we feel that it has merit but does not fully meet PLOS ONE’s publication criteria as it currently stands. Therefore, we invite you to submit a revised version of the manuscript that addresses the points raised during the review process.

We look forward to receiving your revised manuscript.

Kind regards,

Raman Meenakshi Sundaram, Ph.D.

Academic Editor

PLOS ONE

2.We suggest you thoroughly copyedit your manuscript for language usage, spelling, and grammar. If you do not know anyone who can help you do this, you may wish to consider employing a professional scientific editing service.  

 [The funders had no role in study design, data collection and analysis, decision to publish, or preparation of the manuscript.].

4.PLOS ONE now requires that authors provide the original uncropped and unadjusted images underlying all blot or gel results reported in a submission’s figures or Supporting Information files. This policy and the journal’s other requirements for blot/gel reporting and figure preparation are described in detail at https://journals.plos.org/plosone/s/figures#loc-blot-and-gel-reporting-requirements and https://journals.plos.org/plosone/s/figures#loc-preparing-figures-from-image-files. When you submit your revised manuscript, please ensure that your figures adhere fully to these guidelines and provide the original underlying images for all blot or gel data reported in your submission. See the following link for instructions on providing the original image data: https://journals.plos.org/plosone/s/figures#loc-original-images-for-blots-and-gels.

Reviewers' comments:

Reviewer's Responses to Questions

**Comments to the Author**

1. Is the manuscript technically sound, and do the data support the conclusions?

Reviewer #1: Yes

Reviewer #2: Partly

2. Has the statistical analysis been performed appropriately and rigorously? 

Reviewer #1: Yes

Reviewer #2: Yes

3. Have the authors made all data underlying the findings in their manuscript fully available?

Reviewer #1: Yes

Reviewer #2: No

4. Is the manuscript presented in an intelligible fashion and written in standard English?

Reviewer #1: Yes

Reviewer #2: No

5. Review Comments to the Author

Reviewer #1: BLB is an agronomically important bacterial disease that causes significant yield loss in rice worldwide. The OsSWEET14 gene is a known target of Asian Xoo strains and the authors have tried to verify if that also served as a susceptibility factor for Vietnamese Xoo strains using CRISPR/Cas9 mediated genome editing. Though some SWEET genes have already been targeted by different research groups, the authors have edited the promoter of a popular Vietnamese rice cultivar and have challenged it against two Vietnamese Xoo strains. The experiments have been done and presented logically here. Authors have reported a deletion of six-bases in the promoter region of OsSWEET14 gene that resulted in mutants, which were resistant to one of the Xoo strains tested. However, they were unable to verify what is the minimum number of nucleotide deletion required to get high level resistance. If it is possible for authors to do further experiments to verify this, it will strengthen the manuscript.

In many cases, the authors have mentioned the experimental method in the results section and those should be moved back to materials and methods section. For example, Ln 228-232 can be moved to material and methods section.

Ln 36: The plants have been numbered as x.y.z. Commonly, plants are numbered as x-y-z format where x, y and z represent the offspring number of the plant in subsequent generations. I strongly suggest renaming the plants for easy understanding and to follow the convention.

Ln 159: cite appropriate reference for selecting OsEF1α as reference gene for RT-PCR

Ln 303: Significant difference is clearly visible but no asterisks in Fig. 3B

Ln 319: use lower-case or upper-case letters for all means that don’t vary significantly.

Ln 325-327: Gives an impression that only three studies have been reported for rice improvement, which is not true. Please modify the statement!

Ln 367: Authors assume that limited modifications in promoter regions might not have affected the normal expression of SWEET genes in contrast to KO or silenced lines but they must prove it by using RT-PCR or QRT-PCR before the manuscript can be accepted for publication.

Ln 393: Authors claim that probably OsSWEET14 is the only S gene target of tested Xoo strain VXO_11 without testing other SWEET genes. The lesion observed is very less but that is not the enough evidence support this statement. Either they must test the EBEs related to other genes or should remove this statement.

Additionally, there are a lot of typos in the manuscript. I have mentioned few of those below, but the manuscript needs to be verified thoroughly for such grammatical errors.

Ln 60: ration should read as ratio

Ln 67: recessive – delete?

Ln 72: obtained?

Ln 82: World should read as world

Ln 97: conferred/ confer?

Ln 97: Is this new identification or verification?

Ln 220: OsSWEET14 2 day post-infiltration can be written as OsSWEET14 two day post-infiltration

Ln 384: very broad spectrum can be written as broad spectrum

Reviewer #2: Review comment on the manuscript entitled “Improved bacterial leaf blight disease resistance in the major elite Vietnamese rice cultivar TBR225 via editing of the OsSWEET14 promoter” (PONE-D-20-33201)

General Comment: Present manuscript deals with the development of bacterial blight resistant rice lines (TBR225) using CRISPR/Cas9 mediated genome editing tool. Authors have targeted AvrXa7/PthXo3 EBE sequence of SWEET 14 gene promoter. Using CRISPR/Cas9 mediated gene editing system, authors have generated TBR225 mutant lines with disrupted AvrXa7/PthXo3 effector binding element (EBE). One of the mutant lines showed significant resistance to bacterial leaf blight pathogen. Analysis of T0 and T1 transgenic rice plants showed stable inheritance of mutants to the next generation. Agronomic evaluation has also been performed showing no phenotypic difference with the wild type. Overall, the work is interesting, but the manuscript is poorly written with numerous grammatical mistakes/typos/sentence construction errors (although I have mentioned a few below). However, I find several major issues with the manuscript. Please find my specific comments below. They must be addressed satisfactorily before it can be accepted for publication.

Major:

1. The authors have a similar paper published in Vietnamese (http://www.tapchikhoahocnongnghiep.vn/uploads/news/2019_01/7_1.pdf) . I could understand authors have designed 3 guide RNAs in that published report. The full paper is not accessible for a detail verification. Authors need to clarify what additional work they have done in this study.

2. Abstract: “In this study, we proved that the expression of TBR225 OsSWEET14 was induced by the infection of two representative Vietnamese Xoo VXO_11 and VXO_15 strains”. This claim is not supported by the results obtained in the study. Authors have studied the expression of only one member (SWEET14) of SWEET gene family. The data is not sufficient to claim that the two strains induce only SWEET14. Additional SWEET genes need to be included to get a complete picture.

3. I am wondering if Vietnamese Xoo VXO_11 and VXO_15 strains are known to secrete AvrXa7/PthXo3 from any earlier studies. If not, then how the authors have selected SWEET14 for expression analysis and then editing the EBEs? How the authors hypothesized that SWEET14 is the probable target S gene for Xoo VXO_11 and VXO_15 strains?

4. After the initial expression analysis, authors have set out to edit the EBE sequence in the SWEET14 promoter. There are 4 different known EBEs (TalC, TalF, AvrXa7, and PthXo3) located in the promoter. The guide RNAs designed in the study makes a double strand break in the overlapping EBEs for AvrXa7 and PthXo3. Why the authors preferred to target this EBE over TalC and TalF is not clear from the introduction section. Briefly mention that TalC and TalF are mainly present in African lineages. From the analysis of Oliva et al (2019), “Asian strains had approximately equal numbers of PthXo2 (targeting SWEET13) and PthXo3/AvrXa7 (SWEET14)”. This again raises question, why authors choose to go for only Sweet14?

5. From Figure 1C, it is understood that in some lines (e.g. L1.27), deleted bases comprise of both AvrXa7 and TalF EBEs. It would be interesting to know their disease reactions compared to the lines with only AvrXa7 disrupted.

6. For another line of confirmation, expression analysis of SWEET14 gene from the edited lines (before/after infection) would be a great addition. Authors discussed “This incomplete resistance could result from the partial but still productive recognition of subsequences of the altered EBE by a VXO_15 AvrXa7/PthXo3-like TALE.” This could be simply analysed by expression analysis in the edited line.

7. Line 1.5.7 had 6 bp deletion and showed the best resistance response. Please add the sequence details in Figure 1C.

8. L1.5.7 line was resistant to VXO_11 but moderately to the VXO_15 strain. This indicates VXO_15 might possess additional TALE and induces distinct SWEET. Authors have discussed this in line 399-403. This demands additional experiments, at least analyzing expression of SWEET genes.

9. What is the explant for transformation? Please mention in the material method section.

10. What is the percentage of editing in T0 generation? Please provide a table.

11. Have the authors analyzed off-target editing? If so, please mention here.

12. Line no 368…. Authors said …. “our findings are consistent with the previous work with Oliva et al 2019 ----------- multiple combinations of EBE mutations in the OsSWEET11, OsSWEET13 and OsSWEET14 promoters and did not observe abnormal agronomic traits in Kitaake rice”. This information is not fully correct. Genome edited rice line IR64-106 showed phenotypic differences regarding yield, panicle length and fertility. Consider mentioning it.

13. Please provide an image of genome edited TBR225 mutant rice line with its wild type counterpart to show Phenotypic similarity.

Abstract:

1. Please check if ‘Viet Nam’ or ‘Vietnam’ is correct

2. “OsSWEET14 belongs …. host S gene”. Please split into two sentences to make it easy to follow.

3. “Using CRISPR/Cas9 gene editing system, nine TBR225 mutant lines targeting the AvrXa7/PthXo3 effector binding element (EBE) located on OsSWEET14 promoter region were identified from ten transgenic plants.” Replace ‘targeting’ with “with disrupted AvrXa7/PthXo3……”

4. Line 31-32: “Genotyping analysis of T0 and T1 showed that all mutations were stably inherited to the offspring next generation” Follow the suggestion.

Introduction:

Line 51-53: “TALEs are injected into the plant cell, bind to specific nuclear host gene promoter sequences termed Effector-Binding Elements (EBEs) and induce target gene expression to the benefit of the pathogen”. Modify—Once TALEs are …..cell, they bind to……..”

Line 58-59: “All Xoo strains recurrently target S genes belonging to the SWEET gene family and coding for transmembrane sugar exporter proteins”. Needs proper citation.

Material and Methods:

‘Gene expression analysis’ should go earlier than gRNA design.

Other:

1. Explain the SWEET once in the beginning.

2. Why SWEET14 is not clear? Please briefly mention what makes author to go for SWEET14 promoter editing.

3. Figure legends and tables are misplaced. Please rectify.

4. Page 6: gRNA design: I see there are more than 3 distinct guide options to disrupt the EBE (for PthXo3 and AvrXa7). Why the authors have selected the one depicted in Figure 1a need to be briefly explained.

5. Line 347-349: See the discussion of an earlier publication (https://doi.org/10.1007/s42994-020-00018-x). The authors may cite and take help from the paper to discuss additional mutations in T1 generation.

6. Figure 1c, replace ‘Tal5’ with TalF to make it consistent with the figure legend.

7. line 60: Correct the carbohydrate spelling

8. Line 107. ……the sentence --in Hanoi-Vietnam in 2013 and 2016, respectively). Please omit first bracket ‘)’ at the end of the sentence.

9. Line 144…. Please change ‘base on’ to ‘based on’.

10. Line 125. ‘The obtained vector was the integrity of the inserted fragment was verified by sequencing’. Rectify it like- ‘The clone was verified by sequencing’.

11. Line 142. Explain PSA media

12. Line 351. Please correct. genome, , omit extra comma.

13. Line 384. In the sentence ‘to achieve very broad spectrum of resistance’.. Please delete the word very before broad spectrum.

14. fig.2B. replace the figure labelling OsSWEET to OsSWEET14.

15. Line 249: correct analyzis

16. Line 372-385: very poorly written with poor sentence construction. Make them simpler and grammatically correct.

17. Line 249: “All T1 individuals derived from each of two T0 putative homozygotes..” the highlighted portion is not understood.

18. Line 268: “with 88.9% of the T0 lines generating T-DNA-free”. How this 88.9% has been calculated?

19. Line 228: ‘Northern of Vietnam’ correct it

20. Line 111: replace Prime with Primer.

6. PLOS authors have the option to publish the peer review history of their article (what does this mean?). If published, this will include your full peer review and any attached files.

Reviewer #1: **Yes: **Akshaya Kumar Biswal

Reviewer #2: **Yes: **Kutubuddin Molla

---

## [Author Response · Author response to Decision Letter 0]

26 Dec 2020

We would like to thank the appointed reviewers for their time and efforts assessing our work. The feedback we received was extremely helpful for improving the manuscript and we hope we did a satisfactory job at addressing the various points raised during the reviewing process.

---

## [Decision Letter · Decision Letter 1]

5 Feb 2021

PONE-D-20-33201R1

Improved bacterial leaf blight disease resistance in the major elite Vietnamese rice cultivar TBR225 via editing of the OsSWEET14 promoter

PLOS ONE

Dear Dr. Pham,

Thank you for submitting your manuscript to PLOS ONE. After careful consideration, we feel that it has merit but does not fully meet PLOS ONE’s publication criteria as it currently stands. Therefore, we invite you to submit a revised version of the manuscript that addresses the points raised during the review process.

We look forward to receiving your revised manuscript.

Kind regards,

Raman Meenakshi Sundaram, Ph.D.

Academic Editor

PLOS ONE

Reviewers' comments:

Reviewer's Responses to Questions

**Comments to the Author**

1. If the authors have adequately addressed your comments raised in a previous round of review and you feel that this manuscript is now acceptable for publication, you may indicate that here to bypass the “Comments to the Author” section, enter your conflict of interest statement in the “Confidential to Editor” section, and submit your "Accept" recommendation.

Reviewer #1: (No Response)

Reviewer #2: (No Response)

2. Is the manuscript technically sound, and do the data support the conclusions?

Reviewer #1: Yes

Reviewer #2: No

3. Has the statistical analysis been performed appropriately and rigorously? 

Reviewer #1: Yes

Reviewer #2: Yes

4. Have the authors made all data underlying the findings in their manuscript fully available?

Reviewer #1: Yes

Reviewer #2: Yes

5. Is the manuscript presented in an intelligible fashion and written in standard English?

Reviewer #1: Yes

Reviewer #2: Yes

6. Review Comments to the Author

Reviewer #1: The manuscript has been much improved from the previous version. Though there are still some discrepancies that could have improved the manuscript, authors have adequately justified those. For example, they assume that 6 base pair deletion in the promoter region can make a difference in TALE binding but not in gene expression or induction of expression. While working on a promoter editing, it is generally expected to check the expression pattern of the gene under normal and infested condition after editing. Since they did not find any difference in phenotype and it was probably difficult to conduct the experiment, they have avoided the experiment.

Authors claim in the conclusion that the study uncovered potential diversity of TALEs. Though the discussion made by authors indicate towards it, they have not done any experiment to verify that. Hence the statement must be modified accordingly. Though most of the grammatical errors have been rectified still there are some errors as mentioned below:

Ln 33: 'All examined agronomic traits of three transgene-free T2 lines were not significantly different from those of wild-type TBR225' may read as 'None of the examined agronomic traits of three transgene-free T2 lines were significantly different from those of wild-type TBR225'

Ln 67: recessive resistance? – I pointed it earlier, but the explanation was more confusing and must be addressed

Ln381: The single nucleotide mutation was observed only in two of nine plants. So the type of mutation should be only insertion or deletion but not single nucleotide insertion or deletion.

Reviewer #2: Response to manuscript PONE-D-20-33201R1

1. The reply to my comment 3 is not satisfactory scientifically. If authors need to back their gene selection in a rational way, they should cite earlier paper that described Asian strains target SWEET14 or SWEET13. For easy reference see below my comment and author’s response-

My original comment 3: I am wondering if Vietnamese Xoo VXO_11 and VXO_15 strains are known to secrete AvrXa7/PthXo3 from any earlier studies. If not, then how the authors have selected SWEET14 for expression analysis and then editing the EBEs? How the authors hypothesized that SWEET14 is the probable target S gene for Xoo VXO_11 and VXO_15 strains?

Authors replied: As thoughtfully pointed out by the reviewer below, based on previous studies, we knew that Asian strains tend to target either OsSWEET14 or OsSWEET13. So, we just tested OsSWEET14 induction and were very lucky it turned to be the good choice.

2. Figure caption: S4: Explain a bit more about TALVEZ scoring for making it easy for the readers.

3. Similarly, reply to my original comment 6 is not satisfactory. It is not understood why authors are reluctant to perform expression analysis. This experiment needs to be done, otherwise the manuscript looks like a substandard one.

Original comment 6: For another line of confirmation, expression analysis of SWEET14 gene from the edited lines (before/after infection) would be a great addition. Authors discussed “This incomplete resistance could result from the partial but still productive recognition of subsequences of the altered EBE by a VXO_15 AvrXa7/PthXo3-like TALE.” This could be simply analysed by expression analysis in the edited line.

Authors replied: We agree that examining OsSWEET14 expression in the edited lines would help decide between possible explanations for the partial resistance phenotype against

VX0_15. As described in our reply to Reviewer 1's comment #1, we however believe this is beyond the scope of the core results of our study. To tackle this issue, we are in the process of generating the resources to obtain a good vision of the tal genes content of some VXO strains (including VXO_11 and VXO_15). This and the suggested expression assays will be part of a follow up study focusing on the mechanisms explaining these phenotypes.

4. Authors have not performed off-target analysis even for revised manuscript. Which is a standard practice for performing CRISPR-Cas9 experiment. Authors used a single guide and analyzing off-targets for a single guide is an easy task.

5. Authors have not taken care of the following original comment in their discussion in the revised manuscript. Line number 382-390 in the revised manuscript.

Original comment: other 5: Line 347-349: See the discussion of an earlier publication (https://doi.org/10.1007/s42994-020-00018-x). The authors may cite and take help from the paper to discuss additional mutations in T1 generation.

In the Page 116 of the suggested paper, it is discussed “Plants descendent from mutants generated by active Cas9 are prone to further rounds of editing until the PAM and seed region of protospacer are destroyed by editing.” Please also discuss your result in this line.

7. PLOS authors have the option to publish the peer review history of their article (what does this mean?). If published, this will include your full peer review and any attached files.

Reviewer #1: No

Reviewer #2: No

---

## [Author Response · Author response to Decision Letter 1]

6 May 2021

We would like to thank you for your time and efforts handling and assessing our work. We hope our modifications of the initial manuscript address the concerns raised by the journal and reviewers.

---

## [Decision Letter · Decision Letter 2]

15 Jun 2021

PONE-D-20-33201R2

Improved bacterial leaf blight disease resistance in the major elite Vietnamese rice cultivar TBR225 via editing of the OsSWEET14 promoter

PLOS ONE

Dear Dr. Pham,

Thank you for submitting your manuscript to PLOS ONE. After careful consideration, we feel that it has merit but does not fully meet PLOS ONE’s publication criteria as it currently stands. Therefore, we invite you to submit a revised version of the manuscript that addresses the points raised during the review process.

We look forward to receiving your revised manuscript.

Kind regards,

Raman Meenakshi Sundaram, Ph.D.

Academic Editor

PLOS ONE

Journal Requirements:

Additional Editor Comments (if provided):

In view of comments of the reviewers, I recommend the manuscript for a minor revision

Reviewers' comments:

Reviewer's Responses to Questions

**Comments to the Author**

1. If the authors have adequately addressed your comments raised in a previous round of review and you feel that this manuscript is now acceptable for publication, you may indicate that here to bypass the “Comments to the Author” section, enter your conflict of interest statement in the “Confidential to Editor” section, and submit your "Accept" recommendation.

Reviewer #1: All comments have been addressed

Reviewer #2: (No Response)

2. Is the manuscript technically sound, and do the data support the conclusions?

Reviewer #1: Yes

Reviewer #2: Yes

3. Has the statistical analysis been performed appropriately and rigorously? 

Reviewer #1: Yes

Reviewer #2: Yes

4. Have the authors made all data underlying the findings in their manuscript fully available?

Reviewer #1: Yes

Reviewer #2: Yes

5. Is the manuscript presented in an intelligible fashion and written in standard English?

Reviewer #1: Yes

Reviewer #2: Yes

6. Review Comments to the Author

Reviewer #1: The authors have edited the promoter region of OsSWEET14 gene but not the gene itself. This is a perfect approach since editing of the gene itself might have negative effect on seed setting as demonstrated elsewhere in RNA silencing experiments. This was also discussed by the authors in the “Discussion” section. I wish authors discuss this in the introduction to make the concept clear why they selected to edit the promoter region.

Ln 38-40: This conclusive statement of abstract does not match the title. This can be a supplementary statement but not the only statement.

Ln 410-412: The authors have written “In contrast, here, all the T1 plants generated from both heterozygous lines L-21 and L-29, regardless of whether they had a CRISPR/Cas9 T-DNA transgene integrated in their genome, did not show any new mutation possibly because CRISPR/Cas9 T-DNA transgene was no longer functional.” There is no reason why the CRISPR tools will become inactive, and authors did no effort to test this. The statement must be removed since it can give a wrong message.

Reviewer #2: Comments on the manuscript entitles, “Improved bacterial leaf blight disease resistance in the major elite Vietnamese rice cultivar TBR225 via editing of the OsSWEET14 promoter” (PONE-D-20-33201R2)

Although this manuscript is a significantly improved one over its earlier version, the following points need to be addressed before publication. Please find my specific major comments below. Moreover, there are many minor points that require correction and modification. Minor points can be found in the attached annotated PDF file of the manuscript.

1. Why only Sweet 14 expression has been analysed and why authors targeted Sweet 14 and not the others_ need an acceptable justification in the introduction or discussion section.

2. Fig 2B: I am wondering about the level of normal expression of SWEET14 in H2O treated rice samples. Even the authors have mentioned in discussion that normal expression of SWEET genes are necessary for plant development. “It is conceivable that limited modifications in promoter regions do not affect the normal expression of SWEET genes in contrast to KO or silenced lines.”

Authors may have a look at Fig 2B in an earlier publication: 10.1111/pbi.12613, where H2O treated sample showed expression of SWEET14. Why it is different in the current manuscript? Please discuss.

3. Please cite and discuss the observation of A similar article recently published

“Zeng, X., Luo, Y., Vu, N.T.Q. et al. CRISPR/Cas9-mediated mutation of OsSWEET14 in rice cv. Zhonghua11 confers resistance to Xanthomonas oryzae pv. oryzae without yield penalty. BMC Plant Biol 20, 313 (2020). https://doi.org/10.1186/s12870-020-02524-y”

7. PLOS authors have the option to publish the peer review history of their article (what does this mean?). If published, this will include your full peer review and any attached files.

Reviewer #1: **Yes: **Akshaya Kumar Biswal

Reviewer #2: No

---

## [Author Response · Author response to Decision Letter 2]

1 Jul 2021

We would like to thank you for your time and efforts handling and assessing our work. We hope our modifications of the initial manuscript address the concerns raised by the journal and reviewers.

---

## [Decision Letter · Decision Letter 3]

19 Jul 2021

Improved bacterial leaf blight disease resistance in the major elite Vietnamese rice cultivar TBR225 via editing of the OsSWEET14 promoter

PONE-D-20-33201R3

Dear Dr. Pham,

We’re pleased to inform you that your manuscript has been judged scientifically suitable for publication and will be formally accepted for publication once it meets all outstanding technical requirements.

Kind regards,

R M Sundaram, Ph.D.

Academic Editor

PLOS ONE

Additional Editor Comments (optional):

In view of the comments of the reviewer, I recommend that the manuscript may be accepted for publication

Reviewers' comments:

Reviewer's Responses to Questions

**Comments to the Author**

1. If the authors have adequately addressed your comments raised in a previous round of review and you feel that this manuscript is now acceptable for publication, you may indicate that here to bypass the “Comments to the Author” section, enter your conflict of interest statement in the “Confidential to Editor” section, and submit your "Accept" recommendation.

Reviewer #2: All comments have been addressed

2. Is the manuscript technically sound, and do the data support the conclusions?

Reviewer #2: Yes

3. Has the statistical analysis been performed appropriately and rigorously? 

Reviewer #2: Yes

4. Have the authors made all data underlying the findings in their manuscript fully available?

Reviewer #2: Yes

5. Is the manuscript presented in an intelligible fashion and written in standard English?

Reviewer #2: Yes

6. Review Comments to the Author

Reviewer #2: Authors have satisfactorily addressed all my queries and incorporated the suggestions. I recommend publication of the manuscript.

7. PLOS authors have the option to publish the peer review history of their article (what does this mean?). If published, this will include your full peer review and any attached files.

Reviewer #2: **Yes: **Kutubuddin Ali Molla

---

## [Editor Report · Acceptance letter]

31 Aug 2021

PONE-D-20-33201R3 

Improved bacterial leaf blight disease resistance in the major elite Vietnamese rice cultivar TBR225 via editing of the *OsSWEET14* promoter 

Dear Dr. Pham:

I'm pleased to inform you that your manuscript has been deemed suitable for publication in PLOS ONE. Congratulations! Your manuscript is now with our production department. 

Kind regards, 

on behalf of

Dr. R M Sundaram 

Academic Editor

PLOS ONE